# *RAS*-Beppu Classification: A New Recurrence Risk Classification System Incorporating the Beppu Score and *RAS* Status for Colorectal Liver Metastases

**DOI:** 10.3390/cancers17040640

**Published:** 2025-02-14

**Authors:** Takuya Tajiri, Kosuke Mima, Toru Beppu, Hiromitsu Hayashi, Taichi Horino, Yuki Adachi, Katsunori Imai, Toshiro Masuda, Yuji Miyamoto, Masaaki Iwatsuki

**Affiliations:** 1Department of Gastroenterological Surgery, Kumamoto University, Kumamoto 860-8555, Japan; ttajiri005@gmail.com (T.T.); mimakousuke0707@yahoo.co.jp (K.M.); tbeppu@yamaga-mc.jp (T.B.); ja6qpr@gmail.com (T.H.); adatchrugby@gmail.com (Y.A.); katsuimai@hotmail.com (K.I.); masudakumamoto@hotmail.com (T.M.); miyamotoyuji@kumamoto-u.ac.jp (Y.M.); maiwa217@kumamoto-u.ac.jp (M.I.); 2Department of Surgery, Yamaga City Medical Center, Yamaga 861-0593, Japan

**Keywords:** Beppu classification, colorectal liver metastases, JSHBPS nomogram, *RAS*-Beppu classification, *RAS* mutation, recurrence risk stratification

## Abstract

Preoperative recurrence risk stratification for colorectal liver metastases (CRLM) undergoing hepatectomy is essential when designing a treatment strategy. We previously developed a predictive recurrence risk classification, the Beppu classification system, consisting of three risk groups (low, moderate, and high), and found that the *RAS* mutation increased in the low-to moderate- and moderate-to-high-risk groups. We called this new classification the *RAS*-Beppu classification. The new classification showed superior hazard ratios than the previous one in the disease-free survival (DFS) and overall survival (OS). In the multivariate analysis of DFS and OS, the new classification was also statistically significant between each group. *RAS*-Beppu classification using standard parameters is a new and potentially useful recurrence risk classification tool for CRLM.

## 1. Introduction

Colorectal cancer is the third most common malignancy worldwide, and liver metastases develop in 30–50% of patients during the disease [1,2]. Although hepatectomy remains the curative treatment of choice for colorectal liver metastases (CRLM), 50–70% of patients with CRLM develop disease recurrence within 2 years after hepatectomy [3]. Therefore, a preoperative method to accurately predict recurrence is required.

Various recurrence and prognostic scoring systems and nomograms have been reported and validated for liver resection for CRLM [4,5,6,7,8,9,10]. Fong’s clinical risk score (CRS), which consists of five preoperative clinical factors, is the most widely used scoring system worldwide. The Japanese Society of Hepatobiliary and Pancreatic Surgeons (JSHBPS) developed the JSHBPS nomogram to predict disease-free survival (DFS) in CRLM patients undergoing upfront liver resection in 2012 [11]. The annual DFS can be easily calculated using six preoperative clinical parameters: synchronous metastasis (3 points); positive primary lymph nodes (3 points); 2–4 tumors (4 points), ≥5 tumors (9 points); maximum tumor diameter ≥5 cm (2 points); extrahepatic metastasis at liver resection (4 points); and preoperative carbohydrate antigen 19–9 level ≥100 U/mL (4 points). The sum of the points is the Beppu score, and the nomogram has been shown to be sufficiently discriminating in the era of modern chemotherapy [12]. Furthermore, the Beppu classification can be used to stratify patients into three groups of recurrence risk: low risk (≤6 points), moderate risk (7–10 points), and high risk (≥11 points). Hepatectomy alone, hepatectomy and adjuvant chemotherapy, and hepatectomy following preoperative chemotherapy are recommended for patients at low, moderate, and high risk, respectively [13]. Preoperative chemotherapy includes perioperative chemotherapy consisting of preoperative and adjuvant chemotherapy.

*RAS* mutations are also associated with poor survival outcomes in patients with CRLM after liver resection [14]. Prognostic scoring systems, including *RAS* mutation status, have developed and shown excellent prognostic values [1,15,16,17,18]. Among them, the genetic and morphological evaluation (GAME) score [15] and the modified clinical risk score (mCRS) [16], including the *RAS* mutation status, are reported to be useful for both DFS and overall survival (OS). Recently, we reported that the prognosis of CRLM patients can be predicted by combining the Beppu classification and the *RAS* mutation status [19]. Patients with *RAS* mutations in the low- to moderate-risk group were associated with poor prognosis. In contrast, the prognosis was inferior in the high-risk group, irrespective of the *RAS* mutation status.

Based on the Beppu classification system and the *RAS* mutation status, here, we develop a new recurrence risk classification system, named the *RAS*-Beppu classification, and compare its usefulness with the original Beppu classification.

## 2. Materials and Methods

### 2.1. Patients and Study Design

A total of 218 consecutive patients with CRLM were treated with hepatectomy at Kumamoto University Hospital between 2004 and 2020. The Beppu classification was created for patients with initially resectable CRLM [11]; therefore, 45 technically unresectable patients were excluded. Finally, 173 patients with CRLM were enrolled. All patients were histologically confirmed as CRLM and provided written informed consent for participation in the study. The study was approved by the Ethics Review Board of Kumamoto University Hospital (Kumamoto University Hospital approval number: 1291). The study was performed in accordance with the Declaration of Helsinki and the Ethical Principles for Medical and Health Research Involving Human Subjects. Data were collected prospectively and analyzed retrospectively, including demographic, epidemiologic, operative, and oncological data.

### 2.2. Treatment Strategy for CRLM

A multidisciplinary team, including colorectal and hepatobiliary–pancreatic surgeons and medical oncologists, decided whether resection and preoperative chemotherapy were appropriate. For technically resectable CRLM, upfront hepatectomy was undertaken except for patients with poor prognostic factors, such as five or more tumors, size >5 cm, high levels of tumor markers, synchronous disease, resectable extrahepatic metastases, or time to recurrence of <12 months. Patients with insufficient future liver remnants, failure to preserve the Glisson’s capsule or hepatic vein, or unresectable extrahepatic lesions were regarded as technically unresectable and treated with chemotherapy, followed by conversion surgery if CRLM became technically resectable.

Based on the randomized controlled trial reported [20], adjuvant chemotherapy after hepatectomy for CRLM was not uniformly performed and was left to the discretion of the treating physician according to the risk of recurrence and the patient’s wishes.

### 2.3. Assessment of RAS Status

From resected primary colorectal cancer or liver metastases, DNA was extracted from 173 formalin-fixed paraffin-embedded tissue samples and tested for *RAS* mutations. In 86 of the 173 patients, *RAS* status (wild type or mutated) was determined by the MEBGEN RASKET™-B Kit (Medical & Biological Laboratories Co., Ltd., Tokyo, Japan) at a reference laboratory (LSI Medience Corporation and SRL Inc, Tokyo, Japan). The remaining patients were tested with the ddPCR™ KRAS screening multiplex kit and ddPCR™ NRAS screening multiplex kit (Bio-Rad, Hercules, CA, USA). Details are described in a previous paper from our institution [19].

### 2.4. Beppu Classification

A treatment strategy based on the technical and oncological aspects is shown in Appendix A [12]. Resectable patients were divided into three groups by recurrence risk stratification, and the recommended treatment strategy was followed.

### 2.5. RAS-Beppu Classification

Here, we develop a *RAS*-Beppu classification that considers the Beppu score and the *RAS* mutation status (Figure 1). The low-, moderate-, and high-risk groups based on the Beppu classification were identified. We previously investigated the utility of *RAS* status and Beppu score for CRLM patients in low- and moderate-risk groups (19). Only two independent prognostic factors for DFS were identified by multivariable analysis: hazard ratios (HRs) [95% confidence interval (CI)] as 1.93 (1.20–3.10) for *RAS* mutation (vs. wild-type) and 1.78 (1.12–2.89) for Beppu score ≥7 (vs. ≤6). HRs were comparable; therefore, we determined that the impacts of Beppu classification and *RAS* status are nearly equivalent. All patients with wild-type *RAS* were defined as being in the equivalent risk group in the *RAS*-Beppu classification. Patients with *RAS* mutations in the low- and moderate-risk groups were raised to the moderate- and high-risk groups, respectively.

### 2.6. Statistical Analysis

All statistical analyses were performed with JMP Pro statistical software version 16.0.0 (JMP Statistical Discovery LLC 920 SAS Campus Drive Cary, NC, USA). Categorical variables were analyzed using the χ^2^ or Fisher’s exact test, and continuous variables by Student’s *t*-test or the Mann–Whitney test. Survival analyses were performed using the Kaplan–Meier method and were compared with the log-rank test. For univariate and multivariate analyses, cox regression survival analyses were used. Univariable analyses of prognostic factors for DFS and OS were performed by categorical variables. Cutoff values for continuous variables, such as operating time and blood loss, were calculated by drawing Receiver Operating Characteristic (ROC) curves for each, and the value with the highest Area Under the Curve (AUC) was adopted. For multivariate analyses, all factors significantly differenced in the univariate analysis were included. Overall survival was calculated from the date of hepatectomy to the date of death or last follow-up, whereas DFS was calculated from the date of hepatectomy until the date of recurrence or death. To compare the models’ prediction ability, we used ROC analysis using Harrell’s Concordance statistic (C-index). We constructed a Cox proportional hazards model for the ROC analysis using the survival package (version 3.5-5) in R (version 4.3.1). *p* values of <0.05 were considered statistically significant.

## 3. Results

### 3.1. Patient Characteristics and Clinical Outcomes According to the Beppu and RAS-Beppu Classification Systems

The patient characteristics are listed in Table 1. Clinical data included in the JSHBPS nomogram are summarized as the Beppu score. The proportion of RAS mutants in all patients was 37%. According to the *RAS*-Beppu classification, compared with the Beppu classification, there were 17 and 8 fewer patients in the low- and moderate-risk groups, respectively, and 25 more patients in the high-risk group (Appendix A). The proportion of patients with RAS mutations was not different between the three groups. The frequency of preoperative chemotherapy increased as the recurrence risk increased, but the frequency of adjuvant chemotherapy was not different between groups. Perioperative parameters, including operation time, blood loss, transfusion, and major complication rates, were not different between groups.

### 3.2. Disease-Free Survival According to the Beppu and RAS-Beppu Classifications

The DFS curves of the three risk groups based on the Beppu and *RAS*-Beppu classifications were clearly separated, and there was a significant trend for shorter DFS as the risk increased (Figure 2). The five-year DFS rates were 49%, 25%, and 16% in the low-, moderate-, and high-risk groups of the Beppu classification, and 57%, 31%, and 16% in the *RAS*-Beppu classification groups, respectively. Compared with the low-risk group, the HRs for DFS were 1.83 and 2.51 in the Beppu classification moderate- and high-risk groups, respectively, and 2.15 and 3.50 in the *RAS*-Beppu classification groups, respectively.

The results of the univariate analysis of DFS are listed in Appendix A. The Beppu and *RAS*-Beppu classifications were significant prognostic factors in DFS, along with RAS status, preoperative chemotherapy, and resection margin status (Table 2). The significant factors from the univariate analysis were used in the multivariate analysis; RAS mutation status was used only with the Beppu classification. By multivariate analysis, the Beppu and *RAS*-Beppu classifications were independent factors for DFS (Table 2). Beppu classification was selected together with the RAS mutation status. The *RAS*-Beppu classification showed higher HRs than the Beppu classification, and the differences in HRs were significant in all two-group comparisons limited in the *RAS*-Beppu classification (moderate vs. low, high vs. low, and high vs. moderate; Table 2). In addition, Harrell’s C index for the *RAS*-Beppu classification was higher, 0.631, compared to 0.604 for the Beppu classification

### 3.3. Overall Survival According to the Beppu and RAS-Beppu Classifications

By the *RAS*-Beppu classification, there was a significant trend for shorter OS as the risk increased, but the trend was not significant when using the Beppu classification (Figure 3). The five-year OS rates were 68%, 59%, and 44% in the low-, moderate-, and high-risk groups of the Beppu classification and 78%, 59%, and 43% in the *RAS*-Beppu classification groups, respectively. It was difficult to distinguish between the OS in the low-risk and intermediate-risk groups of the two classification systems.

The results of the univariate analysis of OS are listed in Appendix A. The *RAS*-Beppu classification, along with several clinical factors and RAS status, was a significant factor in OS (Appendix A). The *RAS*-Beppu classification was an independent prognostic factor, but the Beppu classification was not. Differences in the HRs in the *RAS*-Beppu classification were significant in the low- vs. high-risk and moderate- vs. high-risk comparisons (Appendix A). Like DFS, Harrell’s C index for the *RAS*-Beppu classification was higher, 0.621, compared to 0.577 for the Beppu classification.

### 3.4. Disease-Free Survival According to the Genetic and Morphological Evaluation (GAME) Score and the Modified Clinical Risk Score (CRS)

Based on the genetic and morphological evaluation (GAME) score [15] and the modified clinical risk score (mCRS) [16], which include the RAS mutation status, we performed univariate and multivariate analyses for DFS using the same cohort. Based on the mCRS, there were initially four groups (0–3); however, there were only two patients in mCRS 3, so we moved these patients into the mCRS 2 group. The DFS curves of the three risk groups from each of the two scoring systems were clearly separated, and there was a significant trend for shorter DFS as the risk increased (Appendix A). The results of the multivariate analysis for DFS are shown for the three classifications that incorporate the RAS status (Appendix A). The *RAS*-Beppu classification and the mCRS were independent predictive factors for DFS but the GAME score was not. The HRs in all two-group comparisons were more significant in the *RAS*-Beppu classification. The C-indexes of mCRS and GAME were 0.558 and 0.594, respectively.

## 4. Discussion

We developed the Beppu classification, consisting of low-, moderate-, and high-recurrence risk groups [11,12,13]. Furthermore, RAS mutation significantly impacted DFS-limited patients with low and moderate risk [19]. The Beppu classification and the novel *RAS*-Beppu classifications were compared in detail. In the current cohort, the DFS curves after hepatectomy were markedly different between low-, moderate-, and high-risk patients when using the Beppu or *RAS*-Beppu classifications [11,12]. There was a noticeable trend toward shorter DFS as risk increased. Therefore, the utility of both classifications is confirmed. We performed univariate analysis for DFS, including various pre-, intra-, and post-operative factors. Notably, the conventional Beppu classification could independently predict DFS in the cohort containing the *RAS* mutation status. Multivariate analysis, including the *RAS*-Beppu classification, showed that the *RAS*-Beppu classification was the only independent predictor for DFS. The *RAS*-Beppu classification HRs were significantly different between all pairs of groups.

*RAS*-Beppu classification is the selection criteria after liver resection; however, this study included 22 (12.7%) patients undergoing ablation in combination with liver resection. It might be helpful for patients undergoing liver resection and ablation; however, it is unknown to be beneficial for patients treated with ablation alone [21].

Using the same cohort, we assessed the utility of the GAME score and mCRS, which include the *RAS* status [15,16]. The *RAS*-Beppu classification and mCRS were independent predictors for DFS; the HRs for the two-group comparisons were most significant for the *RAS*-Beppu classification. These results indicate that the *RAS*-Beppu classification is the most beneficial in stratifying CRLM patients undergoing hepatectomy into risk groups. Additionally, the *RAS*-Beppu classification showed the best value in C-index as a model comparison, compared to the conventional Beppu classification, GAME score, and mCRS.

We also performed an analysis of OS and showed that the *RAS*-Beppu classification was an independent prognostic factor, but the Beppu classification was not. Initially, the JSHBPS nomogram was created to estimate DFS after upfront hepatectomy [11]. It has been reported that DFS is not a surrogate marker for OS [22]. Furthermore, the disease-free interval until recurrence, recurrence sites, and the treatment effect for recurrent diseases can influence OS.

In our earlier report [12], we proposed an optimal treatment strategy based on the Beppu classification for patients with initially resectable CRLM (Appendix A). For low-risk patients, perioperative chemotherapy did not affect long-term outcomes; therefore, upfront hepatectomy without chemotherapy is recommended. For moderate-risk patients, adjuvant chemotherapy is recommended because it leads to a 20% increase in three-year DFS compared with hepatectomy alone. For high-risk patients, chemotherapy prior to hepatectomy leads to increased DFS (HR: 0.44 compared with hepatectomy alone) [13]. We investigated the efficacy of chemotherapy in the three risk-stratification groups in the Beppu and *RAS*-Beppu classifications using our cohort; however, we could not obtain clear results.

Oncological borderline CRLM are considered technically resectable but with a high recurrence rate and poor prognosis after upfront surgery, and several criteria to assist treatment decisions have been developed [23,24]. However, there is no standard definition of borderline CRLM in Japan [25,26]; therefore, we recommend using the high-risk category in the Beppu and *RAS*-Beppu classifications, for which preoperative chemotherapy is recommended.

Circulating tumor DNA (ctDNA) released from cancer cells into the plasma is of interest as a non-invasive tool for the detection of minimal residual disease in patients with non-metastatic and metastatic colorectal cancer [27]. In patients with non-metastatic colorectal cancer, a positive ctDNA after the curative resection of colorectal cancer increases the risk of recurrence, whereas ctDNA before curative resection does not [28,29]. Prospective cohort studies of CRLM (patient numbers <100 in each study) suggest that patients with positive ctDNA after hepatectomy are associated with shorter DFS than those who are negative for ctDNA [30,31]. Further large prospective clinical trials are required to demonstrate the clinical utility of ctDNA analyses in patients with CRLM. In contrast, Beppu and *RAS*-Beppu classifications can be easily estimated from routine laboratory data when diagnosing CRLM.

This study has some limitations. It is a retrospective study with a relatively small number of patients in a single institution. Treatment strategies, including operative procedures, indications of chemotherapy, and chemotherapeutic regimens, have changed over time.

## 5. Conclusions

*RAS*-Beppu classification, which adds patients’ *RAS* status to the Beppu classification, is a new and potentially useful recurrence risk classification tool for CRLM. Validation studies of the Beppu and *RAS*-Beppu classification systems using other large cohorts are required. Furthermore, prospective trials are needed to estimate the utility of preoperative and perioperative chemotherapy for high-risk cohorts in the *RAS*-Beppu classification system.

## Figures and Tables

**Figure 1 cancers-17-00640-f001:**
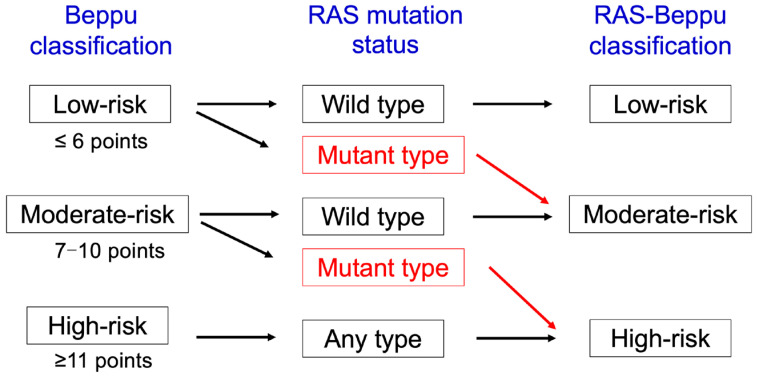
Recurrence risk stratification by the *RAS*-Beppu classification.

**Figure 2 cancers-17-00640-f002:**
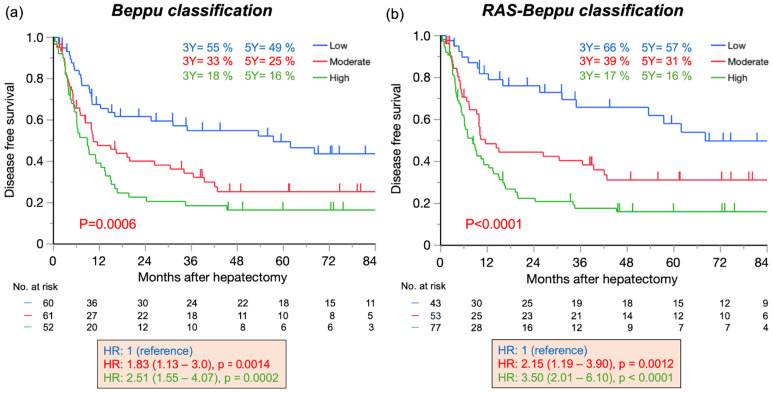
Disease-free survival based on the Beppu (**a**) and *RAS*-Beppu classification (**b**) (n = 173). 3Y: 3-year disease-free survival. 5Y: 5-year disease-free survival. HR: hazard ratio. Beppu classification: low, Beppu score ≤ 6; moderate, 7–10; high, ≥11.

**Figure 3 cancers-17-00640-f003:**
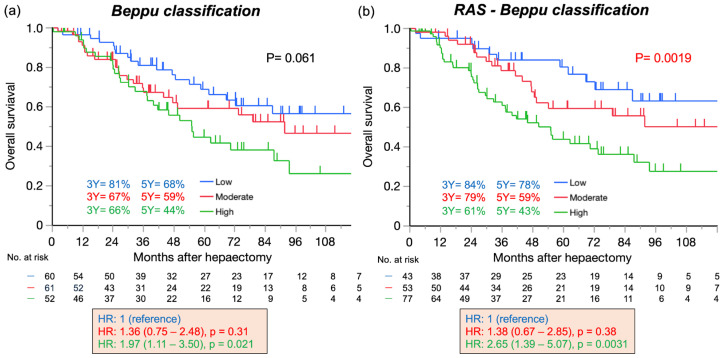
Overall survival based on the Beppu (**a**) and *RAS*-Beppu classification (**b**) (n = 173). 3Y: 3-year disease-free survival. 5Y: 5-year disease-free survival. HR: hazard ratio. Beppu classification: low, Beppu score ≤ 6; moderate, 7–10; high, ≥11.

**Table 1 cancers-17-00640-t001:** Patient characteristics (*n* = 173).

	N = 173
Median age (range)	66 (25–94)
Gender: M/F	60/113
Median BMI (range)	22.7 (14.9–32.5)
Primary tumor sight: Right/Left	40/133
Primary tumor: T 0–2/3–4	19/151
Distribution of liver metastasis: Unilobar/Bilobar	118/54
Median CEA at diagnosis (mg/dL) (range)	9.5 (0.5–3377)
*RAS* mutations (%)	64 (37)
Median Beppu score (range)	7 (0–21)
Chemotherapy (%)	115 (66)
Preoperative	72 (42)
Adjuvant	75 (44)
Operative procedures: Non-AR/AR	109/64
Laparoscopic surgery (%)	55 (32)
TSH or preoperative PVE (%)	5 (3)
Radiofrequency ablation (%)	22 (12.7)
Median operation time (min) (range)	428 (90–1222)
Median intraoperative blood loss (mL) (range)	342 (0–4057)
Transfusion of RBC (%)	12 (7)
Clavien-Dindo class ≥ III (%)	29 (17)
Resection status: R0/R1	121/26

BMI, body mass index; CEA, Carcinoembryonic Antigen; AR, anatomical resection; TSH, two-stage hepatectomy; PVE, portal vein embolization; RBCs, red blood cells; R0, resection margin negative; R1, resection margin positive.

**Table 2 cancers-17-00640-t002:** Univariate and multivariate analysis of disease-free survival (DFS) based on the Beppu and *RAS*-Beppu classification.

DFS			Beppu Classification		*RAS*-Beppu Classification	
	Univariate HR (95% CI)	*p* Value	Multivariate HR (95% CI)	*p* Value	Multivariate HR (95% CI)	*p* Value
*RAS*: mutant/wild	2.25 (1.54–3.29)	<0.0001	2.35 (1.54–3.59)	<0.0001		
Preoperative chemotherapy	1.63 (1.12–2.38)	0.010	1.08 (0.67–1.72)	0.76	1.02 (0.66–1.60)	0.92
Resection margin: R1/R0	1.76 (1.05–2.96)	0.033	1.66 (0.97–2.85)	0.067	1.43 (0.83–2.45)	0.20
Beppu classification		0.0009		0.017		
Moderate/Low	1.83	0.014	1.41	0.21		
	(1.13–3.0)		(0.82–2.46)			
High/Low	2.51	0.0002	2.28	0.005		
	(1.55–4.07)		(1.28–4.05)			
High/Moderate	1.37	0.155	1.61	0.064		
	(0.89–2.11)		(0.97–2.67)			
*RAS*-Beppu classification		<0.0001				0.0002
Moderate/Low	2.15	0.012			2.22	0.015
	(1.19–3.90)				(1.17–4.22)	
High/Low	3.50	<0.0001			3.78	<0.0001
	(2.00–6.10)				(2.00–7.12)	
High/Moderate	1.62	0.025			1.70	0.030
	(1.06–2.48)				(1.05–2.74)	

HR: hazard ratio. R1: resection margin positive. R0: resection margin negative. Beppu classification: low, Beppu score ≤ 6; moderate, 7–10; high, ≥11.

## Data Availability

Data are available upon reasonable request.

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
