# Peer review of "RAS-Beppu Classification: A New Recurrence Risk Classification System Incorporating the Beppu Score and RAS Status for Colorectal Liver Metastases"

_cancers, 2025, doi:10.3390/cancers17040640_

Round 1
Reviewer 1 Report
Comments and Suggestions for Authors
This is an interesting paper with important information.
However, I could not understand why RAS-mutant status should directly connect to grade-shifts from low to moderate and moderate to high.
There is a question whether the impacts of Beppu grade and Ras status is equal.
The authors can make sure whether there are any differences in outcomes (DFS, OS) and backgrounds, when they make the groups of "Beppu-low and RAS mutant", "Beppu-moderate and RAS wild", "Beppu-moderate and RAS mutant", "Beppu-high and RAS wild", and compare them with each other’s.
With those data, they may clarify the importance and the meaning of adding RAS-status to Beppu score.
Author Response
This is an interesting paper with important information.
However, I could not understand why RAS-mutant status should directly connect to grade-shifts from low to moderate and moderate to high.
There is a question whether the impacts of Beppu grade and Ras status is equal.
⇨Thank you for your valuable and informative comment.
We previously investigated the utility of RAS status and Beppu score in the other cohort, including CRLM patients after hepatectomy, with a Beppu score of ≤10 (REF19). There were only two independent prognostic factors for disease-free survival. Hazard ratios (95% confidence interval) in the multivariable analysis showed comparable HRs: 1.93 (1.20–3.10) for RAS mutation (vs. wild-type) and 1.78 (1.12–2.89) for Beppu score ≥7 (vs. ≤6). Therefore, we determined that the impacts of Beppu classification and Ras status are nearly equivalent.
*Inserted table in the Word file.
In contrast, RAS status was not a significant prognostic factor for DFS in patients with a Beppu score of ≥11. Therefore, the definition of the high-risk group was identical for the Beppu and RAS-Beppu classifications.
Of course, the assessment of novel RAS-Beppu classification is not final, and validation studies using other cohorts with many patients are strongly required.
We inserted the following sentences into “2.5 RAS-Beppu classification”.
We previously investigated the utility of RAS status and Beppu score for CRLM patients in low- and moderate-risk groups (19). Only two independent prognostic factors for DFS were identified by multivariable analyses: hazard ratios (HRs) [95% confidence interval (CI)] as 1.93 (1.20–3.10) for RAS mutation (vs. wild-type) and 1.78 (1.12–2.89) for Beppu score ≥7 (vs. ≤6). HRs were comparable; therefore, we determined that the impacts of Beppu classification and Ras status are nearly equivalent.
The authors can make sure whether there are any differences in outcomes (DFS, OS) and backgrounds, when they make the groups of "Beppu-low and RAS mutant", "Beppu-moderate and RAS wild", "Beppu-moderate and RAS mutant", "Beppu-high and RAS wild", and compare them with each other’s.
With those data, they may clarify the importance and the meaning of adding RAS-status to Beppu score.
⇨Thank you for your valuable and informative comment. We compared HRs (95%CI) for DFS in the 6 subgroups. RAS-wild and -mutant patients showed significantly different DFS in the low- and moderate risk groups. Furthermore, DFS was comparable in low-mutant and moderate-wild group, and moderate-mutant and high-wild group. Therefore, we suppose that adding RAS-status to Beppu classification is reasonable.
*Inserted figure & table in the Word file.

Reviewer 2 Report
Comments and Suggestions for Authors
The analysis is robust and the paper is well written. Maybe the paper is too long and some tables/figures could be moved to the supplementary material.
Please add (if any) the censored patients to the KM curves.
What was the AUC of the models? Could be they compared?
Any data on the validation and discrimination of this classification?
Would this classification be applicable also to other treatments for CLRLM, for example radiofrequency ablation? (in this regard cite some other predictive models, for example cite PMID: 27122671)
Author Response
The analysis is robust and the paper is well written. Maybe the paper is too long and some tables/figures could be moved to the supplementary material.
⇨We have now 3 tables and 3 Figures. According to your kind suggestion, we have changed Table 3 to supplementary Table 3. Accordingly, previous supplementary Table 3 has been changed to supplementary Table 4.
Please add (if any) the censored patients to the KM curves.
⇨According to your kind suggestion, we have remade Figures 2 and 3 and Supplementary Figure 2.
What was the AUC of the models? Could be they compared?
⇨Thank you for your informative comment. To compare the models' prediction ability, we used ROC analysis using Harrell's Concordance statistic (C-index). We constructed a Cox proportional hazards model for the ROC analysis using the survival package (version 3.5-5) in R (version 4.3.1).
Below is a summary of the C-index in DFS.
*Inserted the table in the Word files.
According to your valuable comment, we have inserted the following sentences:
In the 2.6 Statistical analysis section.
To compare the models' prediction ability, we used ROC analysis using Harrell's Concordance statistic (C-index). We constructed a Cox proportional hazards model for the ROC analysis using the survival package (version 3.5-5) in R (version 4.3.1).
In the 3.2 Disease-free survival according to the Beppu and RAS-Beppu classifications section.
In addition, Harrell's C index for the RAS-Beppu classification was higher: 0.631, compared to 0.604 for the Beppu classification.
In the 3.3 Overall survival according to the Beppu and RAS-Beppu classifications section.
Like DFS, Harrell's C index for the RAS-Beppu classification was higher: 0.621, compared to 0.577 for the Beppu classification.
In the 3.4 Disease-free survival according to the genetic and morphological evaluation (GAME) score and the modified clinical risk score (CRS) section.
The C-indexes of mCRS and GAME were 0.558 and 0.594, respectively.
In the Discussion section.
Additionally, the RAS-Beppu classification showed the best value in C-index as a model comparison, compared to the conventional Beppu classification, GAME score, and mCRS.
Any data on the validation and discrimination of this classification?
⇨We created the novel RAS-Beppu classification and investigated its usefulness, for the first time. As previously written, the assessment of novel RAS-Beppu classification is not final, and validation studies using other cohorts with many patients are strongly required.
Would this classification be applicable also to other treatments for CLRLM, for example radiofrequency ablation? (in this regard cite some other predictive models, for example cite PMID: 27122671)
⇨According to your valuable comment, we cited the paper (PMID: 27122671), and added the number of patients treated with liver resection with radiofrequency ablation in Table 1, and inserted the following sentences into the Discussion section.
RAS-Beppu classification is the selection criteria after liver resection; however, this study included 22 (12.7%) patients undergoing ablation in combination with liver resection. It might be helpful for patients undergoing liver resection and ablation; however, it is unknown to be beneficial for patients treated with ablation alone [21].

Round 2
Reviewer 1 Report
Comments and Suggestions for Authors
Thank you for your revisions.
Although the authors showed that "DFS was comparable in low-mutant and moderate-wild group, and moderate-mutant and high-wild group", again, are there any differences in OS and backgrounds between the groups of "Beppu-low and RAS mutant", "Beppu-moderate and RAS wild", which are both in RAS-Beppu "moderate"?
Also, are there any differences in those between "Beppu-moderate and RAS mutant" and "Beppu-high", which are both in RAS-Beppu "high"?
The question is "Is there heterogeneity in the RAS-Beppu groups of "moderate" or in "high"?"
Author Response
Comments and Suggestions for Authors
Reviewer 1
Thank you for your revisions.
Although the authors showed that "DFS was comparable in low-mutant and moderate-wild group, and moderate-mutant and high-wild group", again, are there any differences in OS and backgrounds between the groups of "Beppu-low and RAS mutant", "Beppu-moderate and RAS wild", which are both in RAS-Beppu "moderate"? Also, are there any differences in those between "Beppu-moderate and RAS mutant" and "Beppu-high", which are both in RAS-Beppu "high"?
The question is "Is there heterogeneity in the RAS-Beppu groups of "moderate" or in "high"?"
⇨ Thank you for your valuable and informative comment.
We compared HRs (95% CI) for OS in the 6 subgroups. In terms of OS, RAS-wild and mutant patients showed significantly different differences in the moderate-risk but not low-risk groups. Beppu and RAS-Beppu classification was created to predict DFS. As we wrote in the Discussion section, DFS is not always a surrogate marker of OS.
*Inserted the figure & table in the Word file.
We also compared the backgrounds in those 6 subgroups. Because they are grouped according to the Beppu score, there are differences in tumor factors. Other factors are summarized in the following table.
*Inserted the table in the Word file.
・Beppu score (supplementary)
*Inserted the table in the Word file.
Additionally, we compared each 2 groups in terms of DFS, OS.
No statistically significant difference was observed in DFS or OS between the two groups categorized into the same group by the RAS-Beppu classification.
・DFS between “Beppu-low & RAS- mutant” vs. “Beppu-moderate & RAS-wild” and “Beppu-moderate & RAS-mutant” vs. “Beppu-high”.
*Inserted the figure in the Word file.
・OS between “Beppu-low & RAS- mutant” vs. “Beppu-moderate & RAS-wild” and “Beppu-moderate & RAS-mutant” vs. “Beppu-high”.
*Inserted the figure in the Word file.
Furthermore, as the reviewer pointed out, we compared the backgrounds of two groups, "Beppu-low and RAS mutant" and "Beppu-moderate and RAS wild," and "Beppu-moderate and RAS mutant" and "Beppu-high."
・Backgrounds of “Beppu-low & RAS- mutant” vs. “Beppu-moderate & RAS-wild”
*Inserted the table in the Word file.
There were no differences in background between the two groups, "Beppu-low and RAS mutant" and "Beppu-moderate and RAS wild," except for operative time. ・Backgrounds of “Beppu-moderate & RAS-mutant” vs. “Beppu-high”
*Inserted the table in the Word file.
In the "Beppu-moderate and RAS mutant" and "Beppu-high" groups, the “Beppu-high” group had many cases distributed in both lobes, and there were many cases in which perioperative chemotherapy was performed, and RFA was used in combination with liver resection.

Reviewer 2 Report
Comments and Suggestions for Authors
The manuscript is OK. Thank you!
Author Response
We would like to appreciate you for your helpful comments.